# Gonadal Teratomas: A State-of-the-Art Review in Pathology

**DOI:** 10.3390/cancers16132412

**Published:** 2024-06-29

**Authors:** Cecilia Salzillo, Amalia Imparato, Francesco Fortarezza, Sonia Maniglio, Stefano Lucà, Marco La Verde, Gabriella Serio, Andrea Marzullo

**Affiliations:** 1Department of Precision and Regenerative Medicine and Ionian Area, Pathology Unit, University of Bari “Aldo Moro”, 70121 Bari, Italy; sonya1996@hotmail.it (S.M.); gabriella.serio1@uniba.it (G.S.); andrea.marzullo@uniba.it (A.M.); 2Department of Experimental Medicine, PhD Course in Public Health, University of Campania “Luigi Vanvitelli”, 80138 Naples, Italy; stefano.luca@unicampania.it; 3Department of Woman, Child and General and Specialized Surgery, Obstetrics and Gynecology Unit, University of Campania “Luigi Vanvitelli”, 80138 Naples, Italy; amalia.imparato@studenti.unicampania.it (A.I.); marco.laverde@unicampania.it (M.L.V.); 4Surgical Pathology and Cytopathology Unit, University Hospital of Padova, 35121 Padova, Italy; francesco.fortarezza@aopd.veneto.it; 5Department of Mental and Physical Health and Preventive Medicine, University of Campania “Luigi Vanvitelli”, 80138 Naples, Italy

**Keywords:** germ cell tumour, gonadal teratomas, markers, treatment sequencing, quality of life

## Abstract

**Simple Summary:**

Gonadal teratomas are tumours that can be benign or malignant and affect the gonads, specifically the ovaries in women and the testicles in men. They are predominantly young tumours that affect children and adolescents; therefore, early diagnosis is essential to apply the most suitable therapy to preserve fertility. In this article, based on articles found in the literature, we analysed the past, present and future of gonadal teratomas, starting from the description of the possible causes, the distinction between benign and malignant, and consequent diversification of the diagnostic and, therefore, therapeutic procedure. Furthermore, we concluded on the importance of a team that involves various specialists, including the pathologist, for the correct diagnosis and therapy to guarantee the best possible quality of life for the patient.

**Abstract:**

Teratomas are neoplasms arising from germ cells and encompass tissues derived from two or more embryonic germ layers, including ectoderm, mesoderm, and endoderm. These tumours typically localize along the midline or in paramedian positions and can manifest as gonadal (20%) or extragonadal (80%) entities. Although gonadal teratomas are uncommon, they represent the predominant type of gonadal tumour in the paediatric population. They comprise approximately 20–25% of all ovarian tumours in females and about 3–5% of all testicular tumours in males. Ovarian teratomas exhibit a higher incidence in early childhood and adolescence, whereas testicular teratomas are more prevalent during the first three months of life and between the ages of 15 and 19. While the majority of paediatric gonadal teratomas are benign, malignant or mixed variants may also arise, necessitating more aggressive therapeutic interventions.

## 1. Introduction

Teratomas are germ cell tumours which arise from totipotent cells or primitive germ cells and consist of tissues originating from two or more of the three germ layers (ectoderm, mesoderm, and endoderm) [1,2,3]. In rare cases, teratomas are made up of only two layers of germ cells [4].

The etiopathogenesis of teratomas is not yet fully understood and currently three theories have been described in the literature [1,2,3,5]:(1)pluripotent germ cell theory, whereby teratomas are capable of differentiating into any cell of the organism by migrating according to Henson’s node, responsible for localization at the midline and paraxial [6];(2)theory of incomplete twinning or parasitic twin, according to which teratomas derive from the parasitic twin which is incorporated into the dominant twin during embryonic development, responsible for the intracranial, mediastinal, abdominal and sacrococcygeal localization [7];(3)theory of primitive germ cell migration, according to which teratomas migrate from the hindgut towards the genital ridges during embryonic development, responsible for localization to the pelvis and gonads [8].

The genetics of teratomas is complex and varies depending on the type of teratoma and the location. In the literature, deletions of chromosomes 1 and 6 in children and deletions of chromosome 12 in adults are described, as well as N-myc gene expression in immature teratomas [3,9].

In the molecular pathogenesis of germ cell tumours (GCT), primordial germ cells play a fundamental role and in utero undergo a malignant transformation into germ cell neoplasm in situ (GCNS), a precursor of GCT. The primary key events for this transformation are polyploidization and aneuploidization [10].

Molecular features of GCTs include X chromosomal gains and abnormalities of the short arm of chromosome 12, including isochromosome 12p and overrepresentation of chromosome 12p, associated with increased tumour aggressiveness that may influence clinical behaviour [11,12].

GCTs are also characterized by mutations in KIT, KRAS, and CDC27 genes, especially in seminomas, although their precise role is unclear. GCNS contains totipotent cells that can progress to seminoma or embryonal carcinoma, expressing markers such as KIT. These cells undergo epigenetic reprogramming that blocks maturation and promotes malignant transformation, with frequent gain of chromosome 12p [10,13].

Seminomas and non-seminomas display significant differences in the molecular landscape, including DNA methylation patterns, mRNA, miRNA, and proteomic profiles. Seminomas have an unmethylated DNA methylation pattern, while non-seminomas have variable methylation profiles depending on the level of differentiation [10,13].

Teratomas are often solitary but can be associated with other malformations and chromosomal anomalies [2]. In particular, teratoma can be associated with anomalies involving the urogenital system, the central nervous system and congenital dysplasia of the hip [3]. Furthermore, teratomas may be part of the triad of Currarino syndrome as a presacral mass [14] and Klinefelter’s syndrome as a mediastinal teratoma [3], and with rare syndromes such as Proteus, Schinzel–Giedion, Beckwith–Wiedemann, and trisomy 13 and 21 [15,16].

## 2. Classification

Teratomas are classified based on anatomic location and histological features (Table 1).

Anatomically, teratomas are divided into gonadal (20%) or extragonadal (80%) lesions [2]. Gonadal teratomas include ovarian and testicular teratomas. Extragonadal teratomas include:Sacrococcygeal teratoma is the most common congenital tumour with an incidence of 1:35,000–40,000 live births [17], a prevalence of 1:30,000 live births [18] and a female-to-male ratio of 4:1 [19];Mediastinal teratoma with reported frequency percentages ranging from 2.6 to 4 [20,21] and represents approximately 60–70% of mediastinal germ cell tumours [22] and lower incidence in children compared to adults with a ratio of approximately 1:5 [23];Gastric teratoma is very rare and only 102 cases have been described in the literature [24];Retroperitoneal teratoma is rare and accounts for 5% of all childhood teratomas [3] and is even rarer in adults [25];Intracranial teratoma is rare and accounts for a greater than 2–4% of intracranial tumours in children [26];Cervical teratoma is very rare and accounts for only 1.5–5% of all teratomas [27];Other locations such as skin, parotid, vulva, perianal region, spinal canal, umbilical cord and placenta [3].

Histologically, teratomas are classified as mature or immature based on the presence of immature neuroectodermal elements.

A mature teratoma is generally cystic, characterized by the sole presence of well-differentiated tissues deriving from the three germinal layers, such as varying degrees of skin with adnexa, various mucous membranes (digestive, respiratory), exocrine glands (thyroid, pancreas, salivary glands) as well as a variety of mesenchymal tissues (fat, muscle, bone) [28]. This teratoma is the most common type and the diagnosis is confirmed only after surgical excision by histological examination.

Immature teratoma is characterized by undifferentiated elements such as neuroepithelial tissue, as ependymal rosettes or other aggregations of neuroepithelial tissue, which exhibit atypia, a high mitotic count, or hypercellularity, and therefore, may exhibit malignant or recurrent behaviour [2]. The prognosis and therapy are variable depending on the histological grade from 1 to 3 based on the proportion of embryonic tissue [2,28].

Currently, the distinction between mature and immature for testicular teratoma is deemed unnecessary as both are malignant and have no clinical significance according to the Testicular Consultation Group of the International Society of Urological Pathology (ISUP) [29].

## 3. Materials and Methods

Our state-of-the-art review is a comprehensive review of the pathology of gonadal teratoma in the ovary and testis, addressing the topic from pathogenesis to therapy and focusing on the importance of differential diagnosis for the best therapeutic approach.

The review was conducted in PubMed with the following search keywords “Gonadal Teratoma” OR “Ovarian teratoma” OR “Testicular teratoma” OR “Growing teratoma syndrome”, with inclusion criteria for primary studies, secondary studies and English language; additionally, Google Schoolar was used for grey literature.

## 4. Gonadal Teratomas

Gonadal teratomas are tumours that develop from the germ cells of the ovaries (ovarian teratomas) or the testes (testicular teratomas).

Gonadal teratomas are the most frequent congenital embryonic germ cell tumours [30] that originate from primitive pluripotent germ cells [31], usually asymptomatic or with non-specific symptoms such as abdominal pain, distention and compression of the surrounding organs [32] and are often found incidentally following instrumental investigations for another indication [33].

Gonadal tumours are more common in adolescents, in contrast to extragonadal tumours that occur primarily in infants and children, indicating that the location is age-dependent [30].

Although the majority of gonadal teratomas are benign, some may contain malignant elements or become malignant. Treatment usually involves surgical excision, possibly followed by additional therapies such as chemotherapy or radiotherapy, depending on the presence of malignant tissue.

### 4.1. Ovarian Teratomas

Ovarian teratoma (OT) is the most frequent type of germ cell tumour, caused by defective meiosis in the totipotent germ cell, with 46 XX karyotype [5] and less frequently mosaicism, triploidy and trisomy [33].

Histologically, OT is classified into three main categories based on the maturity of the tissues present into mature cystic teratoma (MCT), immature ovarian teratoma (IOT), and monodermal ovarian teratoma (MOT), which includes carcinoid tumours, neural tumours, and ovarian struma [34].

The distinction between the various histotypes is very important as it changes clinical management and prognosis.

#### 4.1.1. Mature Cystic Teratoma

The MCT, also called dermoid cyst, is the most frequent histotype of ovarian teratomas, representing more than 95% of all teratomas and 69% of all germ cell tumours [35], and 20% of all ovarian tumours in adults and 50% of ovarian tumours in children [34].

The average age at diagnosis is 30 years, and it is the most common adnexal mass in pregnant women representing 30.8% of cases [33]. It is benign with a slow growth rate of 1.8 mm/year [34], and in 10% of cases, it involves both ovaries [33].

It is reported in the literature that MCT undergoes malignant transformation in 1–3% with a greater incidence between 40 and 60 years of age [34]; of these, 50% transform into squamous cell carcinoma [36] and, less frequently, malignant melanoma, choriocarcinoma, carcinoids, sarcomas and adenocarcinomas. Additionally, patients with tumours >10 mm are at higher risk of developing malignant transformation [37].

MCT is often asymptomatic and discovered incidentally during imaging examinations for other conditions. When symptomatic, they may present with abdominal pain, distension, or symptoms related to ovarian torsion, haemorrhage, or rupture of the cyst.

The diagnosis of MCT is made by imaging tests such as ultrasonography (US), computed tomography (CT) and magnetic resonance imaging (MRI) and confirmed by histopathology postoperatively [38].

Under US, MCT is characterized by a nonspecific appearance and can present as cystic, solid or complex masses, with echogenic sebaceous materials and calcifications. Specifically, three main types of MCT are seen with US: a cystic lesion with an echogenic tubercle called Rokitansky’s nodule; a diffusely or partially echogenic mass with sound attenuation from sebaceous material and hair; and multiple thin echogenic bands caused by the hairstyle [39]. However, 30% of MCTs may be missed with US due to associated pelvic abnormalities or intrinsic characteristics of MCTs, resulting in a misdiagnosis of malignancy [39].

Very efficient in the diagnosis of MCT is CT which presents with features such as gravity-dependent layers, fat/fluid levels and palm-like protrusions, especially a cauliflower-shaped protrusion which may indicate malignant transformation [39]. Additionally, CT is more efficient than MRI in detecting Rokitansky’s nodule [39].

Under macroscopic examination, the MCT appears as a unilocular cystic mass in most cases, with a size that varies from very small to more than 39 cm, containing sebaceous material that is liquid at body temperature and solid at room temperature and with hair, teeth and bones, and it is more frequently unilateral, involving mainly the right ovary in 72% and bilateral in 12% [34].

Under microscopic examination, the cyst is lined by squamous epithelium with contents characterized by ectodermal tissue such as epithelium and neural tissue, mesodermal tissue such as muscle, fat, bone, and cartilage, and endodermal tissue such as thyroid tissue and gastrointestinal epithelium [34].

The treatment of MCT depends on the age of the patient due to the need to preserve fertility and the risk of malignancy.

In children and young women, when possible, ovarian-sparing surgery called a cystectomy is indicated, with the need for follow-up to exclude recurrence which occurs in 4–5% of cases [40]; however, in women in perimenopause and in menopause, ovariectomy is indicated [38], via laparoscopic or laparotomy.

Laparoscopy is associated with a higher risk of intraperitoneal cyst rupture which increases the risk of chemical peritonitis and formation of peritoneal adhesions, reactive nodules, ascites, and severe abdominal pain requiring repeat surgery, and misdiagnosis may cause iatrogenic loss of malignant cells [41]. In particular, after laparoscopic resection, chemical peritonitis has an incidence of 0.2% in adults and a recurrence of 4.2% in adults and 3–20% in adolescents, associated with factors predictive of recurrence such as young age, cysts ≥ 8 cm and bilateral [41].

In case of malignant transformation such as ovarian carcinoma, therapy consists of surgical removal and chemotherapy [38].

#### 4.1.2. Immature Ovarian Teratoma

IOT is a rare germ cell tumour, which occurs in pure form characterized by the absence of i(12p) or 12p amplification [42] or associated with non-teratomatous germ cell tumours, represents less than 1% of ovarian cancer and is typical of young women, with peak incidence between 16 and 20 years old [43].

Risk factors for IOT are delayed puberty, primary amenorrhea, and racial and ethnic disparities [44].

Clinically, IOT most frequently presents as a painful, unilateral, rapidly enlarging abdominal mass [42]. Sometimes, it is an incidental finding following acute abdominal pain due to tearing or torsion [43].

The diagnosis is made by US, CT and MRI and analysis of serum levels of tumour markers such as AFP, CA 125 and CA 19–9, which have demonstrated specificity and sensitivity in clinical use and are useful for monitoring disease progression [45], and histopathological examination.

Under US, IOT typically presents as solid to cystic heterogeneous masses, generally unilateral; in particular, solid components may contain sebum and calcifications, visible as hyperechoic areas and may show minimal or absent vascularity on Doppler ultrasound. Calcifications in immature teratomas are widespread and not limited to Rokitansky’s nodule and do not cause acoustic shadowing [33].

Under CT and MRI, the IOT is characterized by a large irregular solid component, coarse calcifications and small foci of fat [39].

Macroscopically, IOT appears as a solid and/or cystic mass of variable dimensions, often exceeding 10 cm in diameter; when cut, it appears with solid areas of a friable to compact consistency and of a light grey or yellowish colour, cystic cavities of variable dimensions with sebaceous, mucinous and liquid content, alternating with immature tissues, often with a whitish-grey appearance such as neuroepithelium, the presence of calcified areas, and with areas of necrosis and haemorrhage.

Histologically, the distinguishing features of IOT from MCT are neuroectodermal elements, such as ependymal rosettes or other aggregations of neuroepithelial tissue. Additionally, IOT is characterized by atypia, a high mitotic count or hypercellularity, often interspersed with well-differentiated areas, and may manifest malignant or recurrent behaviour [2].

In OT, it is fundamental for prognosis and therapy to define the grade according to the Norris and O’Connor classification (Table 2) which classifies into four grades based on the characteristics of the neuroepithelium: from completely mature (grade 0) to highly immature (grade 3) and the percentage of the neuroepithelium that varies from less than 10% (grade 1) to more than 50% (grade 3). In contrast, the updated classification (Table 2) divides into low grade (grade 1) and high grade (grade 2 and 3) [2,46,47].

Additionally, small foci of a yolk-sac tumour or embryonal carcinoma may be present which changes the classification to a mixed germ cell tumour [48].

At immunohistochemistry, the neuroepithelium is positive for S100, GFAP and NSE in both the mature and immature components, whereas OCT4 is preferentially expressed in the immature neural tissue of advanced immature teratomas [48].

According to the guidelines of the National Comprehensive Cancer Network (NCCN), a unilateral fertility-sparing adnexectomy associated with peritoneal staging is recommended, while surgery combined with chemotherapy is recommended for advanced stages [49].

If the diagnosis of IOT is made by histological examination without a previous peritoneal biopsy, various therapeutic strategies can be adopted based on the grade of the tumour [49]. In grade I tumours, a second surgery is indicated if peritoneal glial implants are found, as this may indicate the need for chemotherapy. In grade II and III tumours, a second surgery may not be indicated, given that chemotherapy is indicated regardless of the presence of peritoneal implants.

Furthermore, surveillance is recommended for incorrectly staged or unstaged patients and chemotherapy for relapsed patients [43].

#### 4.1.3. Monodermal Ovarian Teratoma

MOT is a rare teratoma consisting mostly or entirely of tissue derived from a single embryonic layer and are classified into ovarian struma, neuroectodermal tumours, or carcinoid tumours [33].

Ovarian struma is more common in women over the age of 40, and 5–10% is malignant. Neuroectodermal tumours are rare, and the primary subtype is the most common, affecting women between 10 and 30 years of age. Carcinoid tumours are more common in postmenopause [33].

Clinically, MOT presents variable symptoms depending on the type of predominant tissue; in particular, the ovarian struma may manifest symptoms of hyperthyroidism, and the carcinoid tumour may present symptoms related to the hormone produced by the tumour.

The diagnosis of MOT is made by imaging tests such as US, MRI and CT, followed by histopathological evaluation of the surgically removed tumour tissue, and tumour markers can be useful in monitoring the disease.

Under CT, ovarian struma may present as complex cystic masses with both cystic and solid components and a smooth exterior; cystic spaces contain areas of hyperintensity and hypointensity, and colloidal material demonstrates high attenuation [50].

Under MRI, carcinoid and neuroectodermal tumours have not been well described, but the majority are solid with heterogeneous enhancement [50].

Macroscopically, monodermal teratomas may appear as solid or cystic masses; the struma ovarii is solid in appearance with greenish-brown areas; the carcinoid tumour is of solid appearance with yellowish or brown areas; and the neuroectodermal tumour has a solid appearance with variable characteristics depending on the tissue differentiation.

Histological examination of the MOT reveals the predominant presence of a highly specialized tissue type; struma ovarii is characterized by thyroid tissue with colloid-containing follicles; the carcinoid tumour is characterized by nests of neuroendocrine cells that are positive for markers such as chromogranin and synaptophysin; the neuroectodermal tumour is characterized by nervous tissue with positivity for neuronal markers such as S100 and NSE.

The main treatment of MOT is surgery, and the need for chemotherapy or radiation therapy depends on the specific type and biological behaviour.

The treatment of ovarian struma varies based on the nature of the tumour: if benign, it is treated with surgical resection; if malignant, it is treated with surgical resection followed by adjuvant therapy with thyroxine and/or radioiodine ablation [51]. The MOT atria are managed surgically, with or without adjuvant chemotherapy [52].

### 4.2. Testicular Teratomas

Testicular teratoma (TT) is a tumour that arises from the germ cells of the testis [53] and represents 50% of mixed germ cell tumours (GCTs) and 3–7% of non-seminomatous testicular germ cell tumours (NSGCT) [54].

According to the current World Health Organization (WHO) classification [55,56], germ cell tumours (GCTs) are classified according to the mode of development, age of onset and subtype, into type I, II and III.

Type I GCTs, also called prepubertal GCTs, develop directly from primordial germ cells and comprise prepubertal teratoma (prTER) [56,57].

Type II GCTs, also called postpubertal, result from a non-invasive precursor lesion such as germ cell neoplasm in situ (GCNIS) and divided seminomas (SEM) and non-seminomatous germ cell tumours (NSGCT) which includes postpubertal teratoma (poTER) [56,57].

The distinction between pre- and post-pubertal GCT is very important since prepubertal GCT usually has benign behaviour with a good prognosis [57].

It is important to differentiate poTER from teratoma with somatic-type transformation (TST), which is rare and defined as a teratoma that develops a distinct secondary component that resembles a somatic malignant neoplasm, with specific prognosis and treatment [54].

#### 4.2.1. Prepubertal Teratoma

PrTER is a rare tumour [58] and is part of the tumours not associated with germ cell neoplasia in situ (non-GCNIS) [56]. It is indolent with no metastatic potential [59], with a diploid chromosome set [57] and an absence of amplification of chromosome 12p [60,61].

PrTER occurs in children aged under 6 years [62] and is less common in adult men after puberty up to age 70 [59]. In postpubertal men, prTER is characterized by monodermal differentiation in most cases [63], and it is possible that many of these teratomas affecting postpubertal patients represent lesions that were not detected during childhood [62].

PrTER is usually discovered incidentally or following nonspecific symptoms, such as abdominal swelling or pain. The diagnosis is confirmed by imaging tests, such as ultrasound, and with macroscopic, histological and molecular examination [54].

US is the gold standard examination for diagnosis and is correlated to the macroscopic morphology of the lesion, whereas MRI is used only in the most complex cases [54].

In addition to traditional imaging methods, it is reported in the literature that Contrast-Enhanced Ultrasound (CEUS) is a useful tool for characterizing testicular masses, with high sensitivity and accuracy, and is effective in distinguishing between neoplastic and non-neoplastic masses and between malign and benign [64].

Under macroscopic examination, prTER appears as a solid or cystic mass, well delimited and often capsulated, with dimensions varying from a few millimetres to several tens of centimetres and with different components, such as hair, adipose tissue, teeth, bones and other tissues, mature or immature [54].

Under histological examination, prTER is characterized by a variety of tissues coming from different germ lines, and it is possible to observe well-differentiated structures, with visible tissue layers that imitate organs such as the epithelium, the lamina propria and the muscularis propria, but any type of tissue such as glands, hair follicles (absent in poTER), bone, cartilage and teeth may be present, along with immature tissues which increase the risk of malignancy and with no cytological atypia [54].

Through molecular analysis, prTER is characterized by the absence of amplification of chromosome 12p, and immunohistochemical markers are positive depending on the tissues derived from the various germ layers.

PrTER is benign and considered cured after complete surgical resection with orchiectomy or local excision when possible, and stressful follow-up programs can be spared [59]. Tumour size is critical in determining whether to perform testicle-sparing or radical procedures, with varying thresholds proposed by different groups. Early guidelines proposed a cutoff of 2.5 cm [57], but more recent data suggest that testicle-sparing surgery is an option for selected tumours larger than 3 or 4 cm [65].

Therefore, the differential diagnosis from poTER is fundamental, especially in adult patients, as the clinical management is completely different. In particular, benign forms of poTER are described in the literature such as the dermoid cyst and the benign non-dermoid teratoma (absence of cutaneous-type adnexal structures), both characterized by the absence of cytological atypia, intratubular neoplasia of the germ cells, unclassified type, significant atrophy tubular/tubular sclerosis, scarred areas, impaired spermatogenesis, microlithiasis and evidence of amplification of chromosome 12p, with a frequent organoid morphology and prominent components of ciliated epithelium and smooth muscle [66].

#### 4.2.2. Postpubertal Teratoma

PoTER derives from GCNIS; specifically, it is part of NSGCT [56], it is a malignant tumour that develops metastases like primary teratoma or other germ cell tumours in 22–37% of cases [54] and is characterized by anomalies of chromosome 12p and by IMP3 expression [67,68].

12p abnormalities cause an excess of DNA from the short arm of chromosome 12 in the form of the 12p-i(12p) isochromosome or a gain in copy number of 12p segments, termed 12p overexpression or 12p gain, respectively [69]. 12p can be detected by karyotyping or by fluorescence in situ hybridization (FISH) in formalin-fixed and paraffin-embedded tissues [67].

IMP3 is expressed physiologically by embryonic tissues. It is responsible for cell migration and early embryogenesis and can be detected in oocytes and granulosa cells in the ovaries, as well as in spermatogonia, spermatocytes and sperm in the testes [70]. According to studies reported in the literature, IMP3 plays a role in the development of GCTs and acts as an oncoprotein that triggers growth, invasion and metastasis in malignant tumours [71]. IMP3 is positive by immunohistochemistry in adult primary and metastatic teratomas [67].

Furthermore, molecular studies report that teratomas show a specific genetic signature, with particular attention to two significantly up-regulated genes such as MMP7 and EGR1 [54,72,73,74,75].

PoTERs represent only 2–6% of testicular cancers, with a mean age of 27.9 years and a range of 20 to 68 years [76].

Clinically, PoTER can present with local symptoms such as painless swelling of the testicle or with systemic symptoms such as weight loss or fever, with about one-third of pure testicular teratomas presenting with advanced disease [77].

PoTER is heterogeneous under US, with a cystic appearance in 28.5%, solid in 21.5% and mixed in 50%, and a thoraco-abdomino-pelvic CT is fundamental to highlighting potential metastasis [76].

Under macroscopic examination, poTER appears as a solid and cystic nodule, with fluid inside the cysts that can be clear, serous, mucinous or keratinizing, and immature teratomas can appear haemorrhagic or necrotic [78].

Under histological examination, poTER is characterized by multiloculated cysts lined by other types of tissue, such as glandular epithelium, and by solid areas formed by parenchymal structure or mesenchymal tissues, and immature tissues, constituted by ectodermal, endodermal or mesenchymal structures and with varying degrees of dysplasia from mild cytologic atypia to evident malignancy. Teratoma may occur in association with other types of germ cell neoplasia, such as yolk-sac tumour or embryonal carcinoma [54].

Through molecular analysis, poTER is characterized by the presence of amplification of chromosome 12p, and immunohistochemical markers are positive depending on the tissues derived from the various germ layers. In particular, glandular tissue exhibits positivity for cytokeratins and epithelial membrane antigen; keratinocytes are positively marked for high molecular weight cytokeratins; neural and adipose tissue shows positivity for S100; muscle fibres express positivity for actin; fibroblastic cells and blood vessels are positive for CD34 [54].

Treatment of poTER usually involves a combination of surgery, chemotherapy, and possibly radiotherapy.

Radical orchiectomy is the gold standard for completely removing the tumour, which may be associated with regional lymphadenectomy. Testicle-sparing surgery is possible for neoplasms that are noninfiltrating and not suspicious for germ cell origin.

After surgery, adjuvant chemotherapy is often administered to eradicate any remaining tumour cells and reduce the risk of recurrence or metastasis. The specific chemotherapy regimen depends on the histological subtype and tumour stage. In some cases, radiotherapy can be used to target residual disease or prevent recurrence in high-risk situations.

Close monitoring and follow-up are essential to evaluate treatment response, detect any relapse, and manage potential long-term side effects.

#### 4.2.3. Teratoma with Somatic-Type Transformation 

TST is defined by the presence of nongerminal somatic components within GCTs and can occur in both primary tumour and metastasis in approximately 3–7% and is confirmed by the presence of the 12p isochromosome like all GCTs [78].

In the literature, different histological types are described as TST, most frequently rhabdomyosarcoma, adenocarcinoma or primitive neuroectodermal tumours (PNET) [79]. Followed by carcinoma of the thyroid gland, signet ring cell adenocarcinoma, papillary renal cell carcinoma, nephroblastoma, non-Hodgkin lymphoma, leukaemia, anaplastic small cell tumour, leiomyosarcoma, liposarcoma, chondrosarcoma, angiosarcoma, malignant tumour of the peripheral nerve sheet, dendritic cell tumour, haemangioendothelioma, carcinoid, glioblastoma, astrocytoma and choroid plexus tumour [78].

Currently, the diagnostic criteria for TST have been updated: measurements are now required in millimetres rather than based on variable visual field sizes. The new criterion establishes a minimum diameter of 5 mm for diagnosis, replacing the previous one based on visible nodules with objective 4 or expansive nodules overlying other GCT elements [80].

The main treatment of TST is surgery; however, those with incomplete resection or metastatic have an unfavourable prognostic outcome [80,81]. Furthermore, TST often displays resistance to chemotherapy, attributable to high expression of multidrug resistance-related protein-1 (MRP1), breast cancer resistance protein (BRCP), and the gene encoding ribonucleoside-diphosphate reductase (RRM1) [78,81].

The therapeutic choice should mirror that used in the standard management of the transformed histological type.

#### 4.2.4. MicroRNAs in Testicular Teratoma

In recent years, microRNAs (miRNAs), particularly microRNA-371a-3p (miR-371a-3p) and microRNA-375 (miR-375), have been the subject of numerous studies as potential biomarkers in testicular GCT (TGCT) [82].

These miRNAs are considered crucial to improve diagnostic and prognostic accuracy in TGCTs, as they can provide information on tumour presence and burden, with particular attention in the differential diagnosis of teratomas, for early identification and evaluation of disease progression [82].

Some studies [83,84] have investigated the efficacy of combined miR-371 and miR-375 as biomarkers to distinguish between teratoma, viable tumour, and necrosis in GCTs, to improve diagnostic accuracy.

Nappi et al. evaluated the use of miR-375 alone or in combination with miR-371 in detecting teratomas and, using two cohorts of patients, highlighted that the plasma miR-371-miR-375 combination was highly accurate in distinguishing teratoma, with area under the curve (AUC) up to 0.95 in the discovery cohort [83].

Kremer et al. investigated the use of miR-371a-3p and miR-375-5p in tissue samples to differentiate between viable GCT and necrosis-fibrosis teratoma. Using a pcRPLND approach on patients with metastatic GCT, they found that the combination of these miRNAs was highly sensitive and specific in differentiating between these conditions, with an AUC of 0.938 [84].

Despite limitations, both studies hypothesize that combined use could represent a significant step forward in more precise diagnosis of GCTs, reducing overtreatment and improving clinical management and, therefore, patients’ quality of life [82].

### 4.3. Growing Teratoma Syndrome

Growing teratoma syndrome (GTS) is a condition in which a teratoma continues to grow during or after first-line chemotherapy treatment despite the elimination of the malignant component and is also characterized by tumour markers within the normal range and histologically by tissue, mature and benign [85]. It is rare with an incidence of 1.9 to 12% [86].

The GTS was first described by Logothetis et al., and in 1977, DiSaia et al. formulated the hypothesis that chemotherapy promotes conversion or retroconversion at the cellular level, or that chemotherapy eradicates all immature chemo-sensitive elements, leaving mature chemo-resistant elements to propagate [87].

Currently, four hypotheses for the development of GTS are described in the literature: (1) chemotherapy inactivates the immature component and prolongs the disease; (2) chemotherapy changes cellular behaviour from malignant to benign; (3) malignant cells show a tendency to spontaneously evolve into benign tissues; and (4) possible unrecognized metastases from immature to mature teratoma [87].

In most cases, GTS occurs within 24 months of completing chemotherapy but is rare after 24 months [86].

Under macroscopic examination, GTS is characterized by both solid and cystic alterations, with sebaceous secretions, teeth, nails and hair follicles inside.

Histologically, GTS is characterized by mature teratomas components such as cartilage, ciliated respiratory-type epithelium, enteric epithelium, and neurogenic tissue, and cystic and solid features may also be present [88]. Furthermore, it is characterized by histological differentiation predominantly with a preference for endoderm, whereas differentiation into ectoderm or mesoderm is rarer [89]. Fundamental for the differential diagnosis between GTS and other teratomas is the absence of immature malignant elements or undifferentiated germ cells.

GTS demonstrates intrinsic resistance to chemotherapy and radiotherapy. Surgery is the first-line treatment, and chemotherapy can shrink the tumour to allow complete removal. Surgery must be timely, preferably via laparotomy or laparoscopy, although the latter may carry a risk of recurrence.

### 4.4. Evaluation of Residual Disease Post-Chemotherapy

Evaluation of post-chemotherapy residual disease in teratoma requires a multidisciplinary approach with imaging, tumour markers, surgery and long-term surveillance, with the goal of complete resection of residual tissue and monitoring to prevent recurrence.

CT and MRI are used to evaluate the size and characteristics of the residual mass and tumour markers, such as alpha-fetoprotein (AFP), human chorionic gonadotropin (HCG) and lactate dehydrogenase (LDH), to determine the response to therapy and relapse.

Specifically, when tumour markers are normal but residual masses >1 cm in size are present, salvage surgery is indicated to remove the residual tissue, which may contain mature teratoma, immature teratoma or residual carcinoma, and dissection of the retroperitoneal lymph nodes (RPLND) is indicated [90,91,92].

Long-term surveillance is performed with periodic imaging and tumour markers to monitor recurrences.

## 5. Conclusions

Our review of the literature highlights that the diagnosis and management of teratomas represent complex challenges even for a team of highly specialized professionals; in particular, the role of the pathologist is fundamental for the differential diagnosis of the various histotypes.

It is crucial to carefully consider the variable biological behaviour of such tumours and the potential unpredictable consequences during the therapeutic pathway to improve patients’ quality of life.

## Figures and Tables

**Table 1 cancers-16-02412-t001:** Classification of teratomas.

Anatomic Location		Histological Features *	
Gonadal teratomas	Extragonadal teratomas	Mature teratomas	Immature teratomas
Ovarian Testicular	Most frequent locations: sacrococcygeal, mediastinal, gastric, retroperitoneal, intracranial and cervical Other less frequent localizations: skin, parotid, vulva, perianal region, spinal canal, umbilical cord and placenta	Well-differentiated tissues, such as varying degrees of skin with adnexa, various mucous membranes, exocrine glands as well as a variety of mesenchymal tissues	Undifferentiated tissues, such as neuroepithelial tissue, which exhibit atypia, a high mitotic count, or hypercellularity

* Currently, the distinction between mature and immature for testicular teratoma is deemed unnecessary as both are malignant and have no clinical significance according to the Testicular Consultation Group of the International Society of Urological Pathology (ISUP).

**Table 2 cancers-16-02412-t002:** Classification of immature ovarian teratomas.

	Classification by Norris and O’Connor	Low- and High-Grade Classification
Grade 0	mature tissue	
Grade 1	≤1 low power field (40×) in a slide	low grade
Grade 2	1–3 low power fields (40×) in a slide	high grade
Grade 3	>3 low power fields (40×) in a slide	high grade

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
