# Peer review of "Gonadal Teratomas: A State-of-the-Art Review in Pathology"

_cancers, 2024, doi:10.3390/cancers16132412_

Round 1
Reviewer 1 Report
Comments and Suggestions for Authors
I read with great interest this review on Gonadal Teratomas.
Below my suggestion:
- Introduction should be revised, please avoid the first lines (28-31). they are not necessary.
- English language should be revised.
- About Testicular teratomas. This part should be enhanced.
- Are there any reliable biomarkers or imaging tools to diagnose Teratoma? MRI ? CEUS? authors should add a paragraph on this part authors may rely on https://doi.org/10.1016/j.ejrad.2008.07.044 and https://doi.org/10.3390/app11198990
Comments on the Quality of English Languagemajor revision required
Author Response
Dear Reviewer,
thank you for the comments and we modified the manuscript as indicated:
Comment 1: Introduction should be revised, please avoid the first lines (28-31). they are not necessary.
Response 1: We removed lines 28 to 31 from the introduction and we rewrote the etiopathogenesis and part concerning the association with malformations and syndromes (highlighted in yellow).
Comment 2: English language should be revised.
Response 2: We asked an expert in the field to review the English text.
Comment 3: About Testicular teratomas. This part should be enhanced.
Response 3: We expanded the section on testicular teratomas by describing imaging techniques (highlighted in blue: lines 344-349), using the suggested reference: https://doi.org/10.3390/app11198990 [65]. We included in the text the reference [67] as a differential diagnosis: lines 370-377 and highlighted in grey.
Comment 4: Are there any reliable biomarkers or imaging tools to diagnose Teratoma? MRI ? CEUS? authors should add a paragraph on this part authors may rely on https://doi.org/10.1016/j.ejrad.2008.07.044 and https://doi.org/10.3390/app11198990.
Response 4: We expanded the section on ovarian teratoma by describing imaging techniques (highlighted in blue: lines 180-191 and 236-237), using the suggested reference: https://doi.org/10.1016/j.ejrad.2008.07.044 [40].
Sincerely
Reviewer 2 Report
Comments and Suggestions for Authors
The authors provided a comprehensive review of the gonadal teratoma pathology in ovary and testis. Review included proposed theories of etiopathogenesis of teratomas, classification of teratomas, pathological features of different types of teratomas in ovary and testis, emphasizing on mature versus immature teratomas with diagnostic criteria, prognosis and treatment strategies, and molecular characteristics of various teratomas.
In testicular teratomas, the authors discussed prepubertal teratoma and postpubertal teratoma, which is significant clinically. The histological diagnosis of prepubertal teratoma, especially in postpubertal men can be challenging. Zhang et al. (Am J Surg Pathol., 37 (2013), pp. 827-835) described dermoid cyst and nondermoid benign teratoma both show “absence of all of the following: cytologic atypia, intratubular germ cell neoplasia, unclassified type, significant tubular atrophy/tubular sclerosis, scarred zones, impaired spermatogenesis, microlithiasis, and evidence of chromosome 12p amplification. Other features include frequent organoid morphology and prominent components of ciliated epithelium and smooth muscle.” WHO 5th edition provided a similar description of prepubertal teratoma. Recommend the authors provide references for paragraph from lines 303 to 308 and explain a bit more detail of the histological findings, especially in postpubertal men due to clinical impact.
The authors mentioned “Laparoscopy is associated with a higher risk of intraperitoneal cyst rupture which increases the risk of chemical peritonitis and adhesion formation, and misdiagnosis may result in iatrogenic leakage of malignant cells”. Recommend include more information to describe incidence, complications, recurrence, etc.
Growing teratoma syndrome was defined as metastatic masses during or after chemotherapy for germ cell tumors, which contain only mature teratoma components, and with normalized tumor markers. Line 416 mentioned undifferentiated mesenchymal spindle-shaped cells. Recommend provide more histological description of the components of the metastasis.
Comments on the Quality of English LanguageMinor editing to express the statement or concept clearly.
Author Response
Dear Reviewer,
thank you for the comments and we modified the manuscript as indicated:
Comment 1: In testicular teratomas, the authors discussed prepubertal teratoma and postpubertal teratoma, which is significant clinically. The histological diagnosis of prepubertal teratoma, especially in postpubertal men can be challenging. Zhang et al. (Am J Surg Pathol., 37 (2013), pp. 827-835) described dermoid cyst and nondermoid benign teratoma both show “absence of all of the following: cytologic atypia, intratubular germ cell neoplasia, unclassified type, significant tubular atrophy/tubular sclerosis, scarred zones, impaired spermatogenesis, microlithiasis, and evidence of chromosome 12p amplification. Other features include frequent organoid morphology and prominent components of ciliated epithelium and smooth muscle.” WHO 5th edition provided a similar description of prepubertal teratoma. Recommend the authors provide references for paragraph from lines 303 to 308 and explain a bit more detail of the histological findings, especially in postpubertal men due to clinical impact.
Response 1: We added in the text the reference suggested [67] as a differential diagnosis: lines 370-377 and highlighted in grey.
Comment 2: The authors mentioned “Laparoscopy is associated with a higher risk of intraperitoneal cyst rupture which increases the risk of chemical peritonitis and adhesion formation, and misdiagnosis may result in iatrogenic leakage of malignant cells”. Recommend include more information to describe incidence, complications, recurrence, etc.
Response 2: We included additional information on post-laparoscopy rupture: lines 207-213 and highlighted in grey.
Comment 3: Growing teratoma syndrome was defined as metastatic masses during or after chemotherapy for germ cell tumors, which contain only mature teratoma components, and with normalized tumor markers. Line 416 mentioned undifferentiated mesenchymal spindle-shaped cells. Recommend provide more histological description of the components of the metastasis.
Response 3: We modified the histological description of the GTS reporting new references: lines 500-505 and highlighted in grey.
Also, we have the English editing done by an expert.
Sincerely
Reviewer 3 Report
Comments and Suggestions for Authors
This is an interesting manuscript on teratomas. The topic has received recent interest in terms of biology, clinical management and classification.
The paper is well written but can be improved:
1. The manuscript must include a short section in which authors explain type of review and literature searching criteria that have been followed to complete the review.
2. Foot note on Table 1 must state that is malignant to avoid misconceptions by the reader.
3. Perhaps, the section on teratoma with somatic-type transformation can be expanded and report most common types, diagnostic criteria and potential therapy. The current WHO 2022 has reviewed the issue and provide now measurements to be applied in practice.
4. Add a section on molecular pathogenesis of teratomas in the context of new developments of GCT, including if possible, a diagram to explain the relationships and relevant molecular differences (chromosome 3?). (please see example Pathologie https://doi.org/10.1007/s00292-023-01264-8)
5. Add a section on the role of microRNAs as noninvasive diagnosis of teratoma. This is a ery relevant and expanding area of research that deserved quotation in a paper like this. Add some recent references pertinent to the issue.
6. Add a short section of evaluation of residual disease after chemotherapy.
Author Response
Dear Reviewer,
thank you for the comments and as indicated:
Comment 1: The manuscript must include a short section in which authors explain type of review and literature searching criteria that have been followed to complete the review.
Response 1: We included the "materials and methods" section: lines 127-133 and highlighted in green.
Comment 2: Foot note on Table 1 must state that is malignant to avoid misconceptions by the reader.
Response 2: In the footnote of table 1 and in the text lines 121-124 (highlighted in green) it is reported "Currently, the distinction between mature and immature for testicular teratoma is deemed unnecessary as both are malignant and have no clinical significance according to the Testicular Consultation Group of the International Society of Urological Pathology (ISUP) [29]". Isn't it explanatory? Is it necessary to write it in a different way?
Comment 3: Perhaps, the section on teratoma with somatic-type transformation can be expanded and report most common types, diagnostic criteria and potential therapy. The current WHO 2022 has reviewed the issue and provide now measurements to be applied in practice.
Response 3: We added the 2022 WHO diagnostic criteria [81]: lines and highlighted in green.
Comment 4: Add a section on molecular pathogenesis of teratomas in the context of new developments of GCT, including if possible, a diagram to explain the relationships and relevant molecular differences (chromosome 3?). (please see example Pathologie https://doi.org/10.1007/s00292-023-01264-8).
Response 4: We added a paragraph on molecular pathogenesis in the introduction by using the one indicated as a reference https://doi.org/10.1007/s00292-023-01264-8 [10] and [13] (lines 59-75 and highlighted in green).
Comment 5: Add a section on the role of microRNAs as noninvasive diagnosis of teratoma. This is a very relevant and expanding area of research that deserved quotation in a paper like this. Add some recent references pertinent to the issue.
Response 5: We added a section on “MicroRNAs in testicular teratomas”: lines 457-479 and highlighted in green.
Comment 6: Add a short section of evaluation of residual disease after chemotherapy.
Response 6: We added a section on “Evaluation of residual disease post-chemotherapy” [91-93]: lines 512-524 and highlighted in green.
Sincerely
Round 2
Reviewer 1 Report
Comments and Suggestions for Authors
The authors addressed my major concerns. I endorse acceptance of the manuscript.